# A Rapid Approach for Identifying Cell Lines Lacking Functional Cytidine Deaminase

**DOI:** 10.3390/ijms26073344

**Published:** 2025-04-03

**Authors:** Anna Ligasová, Markéta Kociánová, Karel Koberna

**Affiliations:** 1Institute of Molecular and Translational Medicine, Faculty of Medicine and Dentistry and Czech Advanced Technology and Research Institute, Palacký University Olomouc, Hněvotínská 5, 779 00 Olomouc, Czech Republic; 2Institute of Molecular and Translational Medicine, Faculty of Medicine and Dentistry, Palacký University Olomouc, Hněvotínská 5, 779 00 Olomouc, Czech Republic

**Keywords:** 5-fluorocytidine deamination, cytidine deaminase, 5-fluorouridine, equilibrative and concentrative nucleoside transporters

## Abstract

CDD plays a pivotal role within the pyrimidine salvage pathway. In this study, a novel, rapid method for the identification of cell lines lacking functional cytidine deaminase was developed. This innovative method utilizes immunocytochemical detection of the product of 5-fluorocytidine deamination, 5-fluorouridine in cellular RNA, enabling the identification of these cells within two hours. The approach employs an anti-bromodeoxyuridine antibody that also specifically binds to 5-fluorouridine and its subsequent detection by a fluorescently labeled antibody. Our results also revealed a strong correlation between the 5-fluorouridine/5-fluorocytidine cytotoxicity ratio and cytidine deaminase content. On the other hand, no correlation was observed between the 5-fluorouridine/5-fluorocytidine cytotoxicity ratio and deoxycytidine monophosphate deaminase content. Similarly, no correlation was observed between this ratio and equilibrative nucleoside transporters 1 or 2. Finally, concentrative nucleoside transporters 1, 2, or 3 also do not correlate with the 5-fluorouridine/5-fluorocytidine cytotoxicity ratio.

## 1. Introduction

Despite the rise of immunotherapy, chemotherapy remains a crucial component of cancer treatment. The efficacy of cytidine analog-based chemotherapeutics is significantly influenced by cellular deaminases, such as cytidine deaminase (CDD, EC 3.5.4.5) and deoxycytidine monophosphate deaminase (dCMP deaminase, DCTD, EC 3.5.4.12). CDD plays a key role in the pyrimidine salvage pathway. The primary role of cytidine deaminase in cellular metabolism is its enzymatic activity as a hydrolase that catalyzes the irreversible hydrolytic deamination of cytidine and deoxycytidine. This reaction converts cytidine to uridine (U) and deoxycytidine to deoxyuridine (dU), respectively. In the absence of CDD, cells synthesize U and dU via de novo pathways [1]. Both enzymes typically inactivate drugs based on cytidine analogs by deaminating their deoxynucleoside or deoxynucleotide forms. Examples of such modified nucleoside drugs include gemcitabine, cytarabine, and decitabine, which are inactivated by deamination [2,3,4,5,6,7].

The mechanisms of action of these drugs vary substantially. Gemcitabine, once incorporated into DNA, triggers premature chain termination after the insertion of another nucleotide triphosphate. This ‘masked’ chain termination prevents DNA repair enzymes from removing gemcitabine, ultimately leading to apoptosis [8,9]. Gemcitabine predominantly induces cell cycle arrest in the S phase, although higher concentrations can cause apoptosis in G1 and G2/M phases [10]. Similarly, cytarabine primarily targets cells undergoing DNA synthesis in the S phase, inducing DNA fragmentation and chain termination [11]. The mechanism of action of decitabine is dose-dependent. At high doses, decitabine forms DNA adducts, resulting in DNA synthesis arrest and cytotoxicity. At low doses, it alters gene expression profiles, promoting differentiation, reducing proliferation, and/or inducing apoptosis [12,13].

Conversely, another drug, capecitabine, requires CDD activity for its activation. Capecitabine is initially converted to 5′-deoxy-5-fluorocytidine by carboxylesterase and subsequently to 5′-deoxy-5-fluorouridine by CDD, primarily in the liver [14,15]. Finally, 5′-deoxy-5-fluorouridine is converted to 5-fluorouracil by thymidine phosphorylase, an enzyme exhibiting increased activity in tumor tissue [16,17]. After its conversion to 5-fluorouracil, capecitabine is further metabolized to 5-fluoro-2′-deoxyuridine monophosphate (FdUMP). FdUMP inhibits thymidylate synthase, disrupting the synthesis of thymidine triphosphate. This depletion of thymidine triphosphate impairs DNA replication, ultimately triggering apoptosis in proliferating cancer cells [18,19]. Therefore, the content/activity of CDD and DCTD plays an important role in the impact of the treatment by all these drugs.

CDD and DCTD protein content can be evaluated by Western blot analysis and gene expression levels can be determined using RT-qPCR. These methods necessitate cell lysis and analysis of cell lysates, extending the experimental timeline.

In a recent study, we introduced a rapid method for identifying cell lines deficient in dCMP deaminase function [20]. This method relied on the incorporation of 5-ethynyl-2′-deoxycytidine (EdC) and its subsequent conversion to 5-ethynyl-2′-deoxyuridine (EdU). Building upon this approach, we investigated the potential of 5-fluorocytidine as a cost-effective and efficient alternative for CDD-deficient cell line identification.

## 2. Results and Discussion

### 2.1. Method Overview

In our previous work [20], we developed a method to identify DCTD-deficient cell lines by monitoring EdU incorporation into DNA in cells cultured in medium containing EdC. Since EdC is also deaminated by CDD, co-treatment with the CDD inhibitor tetrahydrouridine was necessary to ensure the specific labeling of cells exhibiting DCTD activity.

In this study, we investigated the feasibility of using 5-fluorocytidine (FC) to identify CDD-deficient cell lines. The metabolic pathways of FC (Figure 1) demonstrate that utilizing FC eliminates the need to inhibit DCTD activity when monitoring the incorporation of its deaminated product, 5-fluorouridine (FU), into RNA. In addition, we also determined whether the developed approach allows for the comparison of different cell lines in terms of CDD content and activity by performing a calibration to quantify the production of FU from FC.

Initially, cells were incubated with various FU concentrations or 10 μM of FC for one hour. During this incubation, cells take up both FC and FU. Depending on the uptake, CDD activity, and FU phosphorylation, FC is transported into the cells and deaminated by CDD, and the product of this deamination (FU) is subsequently phosphorylated to FU-triphosphate and incorporated into cellular RNA. Notably, in the absence of or with negligible CDD activity, little to no FU should be incorporated into RNA.

Subsequently, samples were fixed with 2% formaldehyde and permeabilized with 0.2% Triton X-100. After permeabilization, cells were incubated with an anti-bromodeoxyuridine (anti-BrdU) antibody, which also recognizes FU, followed by washing and incubation with an Alexa Fluor 488-coupled secondary antibody that recognizes the anti-BrdU antibody. DNA was co-stained with DAPI. The fluorescence images were recorded by a fluorescence microscope. The DAPI-positive areas were defined, and the Alexa Fluor 488 signal was measured within these areas. Finally, the Alexa Fluor 488 signal in these areas was calculated. The signal was corrected by subtracting the background signal obtained from cells incubated in FU- and FC-free medium.

A calibration curve was constructed using the signals from cells incubated with various concentrations of FU. FU production from 10 μM of FC was estimated using GraphPad Prism 6 software by interpolating unknown values from the standard curve (standard four parameter logistic non-linear regression). Figure 2 exemplifies the non-linear regression for HeLa cells, including the interpolated value for 10 μM of FC. The calculated value of FU production should be proportional to CDD activity and to the FC/FU transport ratio and inversely proportional to FC phosphorylation.

Besides formaldehyde fixation and Triton X-100 permeabilization, protocols based on the successive incubation of cells in 2% formaldehyde and 100% methanol, incubation in 100% methanol, and incubation in 70% ethanol were tested. The FU-RNA signal/background ratio was calculated for each protocol. In these experiments, HeLa cells were incubated with 0 (for background measurement) or 10 µM of FU for one hour. It is obvious that formaldehyde fixation followed by triton permeabilization provided the best signal/background ratio (Appendix A). In this respect, we used formaldehyde fixation and Triton X-100 permeabilization in the following experiments.

### 2.2. Method Validation and Optimization

We used NCI-H2009, A549, HeLa, HepG2, hTERT RPE-1, and IMR-90 cell lines to validate the method. These include both cell lines with (NCI-H2009, A549, HeLa, HepG2) and without CDD expression (hTERT RPE-1, and IMR-90). NCI-H2009, A549, HeLa, and HepG2 cells were chosen as they exhibit a very high correlation between CDD activity and content (Pearson correlation coefficient R  =  0.99, *p*  <  0.001) [20]. The results showed that only NCI-H2009, A549, HeLa, and HepG2 produced FU-labeled RNA (Figure 3a). This finding was in complete agreement with the absence of CDD in the hTERT RPE-1 and IMR-90 cell lines (Figure 3b; adapted from Figure 4 of [20]). Furthermore, it was evident that even a very high amount of DCTD did not result in the production of FU. These results confirmed our presumption that DCTD inhibition is not required when FC is used for the evaluation of CDD activity.

Our analysis of FU production and CDD content reveals that while this method can identify CDD-deficient cell lines, it is not suitable for quantifying CDD content in cell lines expressing CDD. Specifically, we found no significant correlation between FU production and CDD content in NCI-H2009, A549, HeLa, and HepG2 cells (Pearson coefficient = 0.8105, *p*-value = 0.19).

Apparently, for qualitatively determining the presence or absence of CDD, the method can be significantly simplified. We tested a simplified arrangement involving the incubation of cells in culture medium containing 50 µM of FU, 50 µM of FC, or without FU/FC (control cells). Mean FU-RNA signal values were measured from at least 1000 cells in each experiment, and signals from FU- and FC-treated cells were compared to control cells using the Mann–Whitney U test (see Table 1). This analysis revealed that only the hTERT RPE-1 and IMR-90 cell lines exhibit CDD deficiency.

### 2.3. The Amount of CDD Strongly Correlated with the FU/FC Cytotoxicity Ratio

Although neither FU nor FC are used as drugs per se, FU is a metabolite of 5-fluorouracil (5-FUr), a widely used chemotherapeutic agent for colorectal cancer [21]. 5-FUr exerts its cytotoxic effects primarily through the inhibition of thymidylate synthase, leading to impaired DNA synthesis. While the incorporation of 5-FUr into DNA and RNA has been proposed as additional mechanisms of action, their precise roles remain less clear [22].

Our data demonstrated that cancer cell lines exhibited lower susceptibility to FU compared with primary cell lines (hTERT RPE-1, IMR-90, *p*-value = 0.028). Considering the extensive incorporation of FU into RNA after phosphorylation, this finding suggests that while RNA incorporation contributes to FU toxicity in cancer cells, it likely plays an even greater role in the adverse effects observed in healthy tissues during cancer therapy.

Analysis of FC and FU cytotoxicity revealed a significant correlation between CDD content and the FU_IC50/FC_IC50 ratio (FU/FC toxicity ratio; Pearson correlation coefficient = 0.89, *p*-value = 0.02, Spearman correlation coefficient = 0.98561, *p*-value = 0.0003). This correlation was nearly perfect when analyzing only cell lines expressing CDD (Pearson correlation coefficient = 0.99 and *p*-value < 0.01, Spearman correlation coefficient = 1, *p*-value < 0.0001). This high correlation strongly emphasizes the crucial role of CDD in FC toxicity and suggests that the primary toxicity of FC can be attributed to its deamination product, FU.

Correlations between the FU/FC toxicity ratio and the expression levels of relevant nucleoside transporters and DCTD were also analyzed (Table 2). However, no significant correlations were found.

## 3. Materials and Methods

### 3.1. Cell Cultures

The following cells were used in this study: human cancer cells HeLa (cervix, adenocarcinoma; ATCC, Manassas, VA, USA; CCL-2), NCI-H2009 (lung, adenocarcinoma, [23]), and A549 (lung, carcinoma, [20])—the latter two cell lines were a gift from Dr. Marián Hajdúch, Institute of Molecular and Translational Medicine, Olomouc; IMR-90 (diploid fibroblasts, lung, ATCC, Manassas, VA, USA; CCL-186); HepG2 (liver, hepatocellular carcinoma; [24]); and non-cancer hTERT RPE-1 (diploid, pigmented epithelium; retina hTERT-immortalized; [25])—the latter two cell lines were a gift from Dr. David Staněk, Institute of Molecular Genetics CAS, Prague.

The HeLa, A549, NCI-H2009, and hTERT RPE-1 cells were cultivated in Dulbecco’s modified Eagle’s medium (DMEM) supplemented with 10% fetal bovine serum, 3.7 g/L of sodium bicarbonate, and 50 µg/mL of gentamicin. HepG2 and IMR-90 cells were cultivated in Eagle’s minimum essential medium (EMEM) supplemented with 20% fetal bovine serum, 3.7 g/L of sodium bicarbonate, and 50 µg/mL of gentamicin. The cells were cultivated at 37 °C in a humidified atmosphere containing 5% CO_2_. All cell lines were regularly tested for mycoplasma contamination by PCR and enzymatic detection [26].

### 3.2. Tested Fixation Protocols

Initially, several fixation and permeabilization protocols were tested. In these experiments, HeLa cells cultivated on circular coverslips (12 mm in diameter) were incubated with or without 50 µM of FU for one hour at 37 °C in a humidified atmosphere containing 5% CO_2_. Then, the cells were quickly rinsed with 1 × PBS buffer (three times) and fixed with either 2% formaldehyde (10 min, RT), ice-cold 100% methanol (5 min, −20 °C), or 70% ethanol (10 min, RT). After formaldehyde fixation, the cells were washed with 1 × PBS (three times), permeabilized with either 0.2% Triton X-100 or 100% methanol (both 10 min, RT), and washed again with 1 × PBS (three times). The methanol-fixed cells were washed with 1 × PBS buffer (three times). The ethanol-fixed cells were air-dried after the ethanol solution was removed and then washed with 1 × PBS.

After fixation and permeabilization, all samples were washed with TNTB buffer. Then, the steps were the same as in Section 3.3.

### 3.3. FC and FU Treatment and FU Detection

The cells were cultivated onto circular coverslips (12 mm in diameter) placed in eight Petri dishes. The cells were seeded one day before the experiments. The cell confluency was around 80%. On the day of the experiment, the following FU concentrations were added to the culture medium of each Petri dish: 0; 0.4; 2; 10; 50; 250; 1250 µM. 10 µM of FC was added to the last eighth Petri dish. Cells were cultivated for one hour at 37 °C in a humidified atmosphere containing 5% CO_2_. After incubation, the cells were quickly rinsed three times with 1 × PBS buffer and fixed with 2% formaldehyde in 1 × PBS (10 min, room temperature (RT)). Then, the samples were rinsed with 1 × PBS and permeabilized with 0.2% Triton X-100 (10 min, RT). Afterwards, the cells were washed three times for 10 min each with the TNTB buffer (25 mM Tris-HCl, pH ~ 7.5; 150 mM NaCl, 1% BSA, 50 mM glycine, 0.1% Tween 20, and 0.02% sodium azide) and incubated with the primary anti-BrdU antibody recognizing FU as well in TNTB (1:300, clone BU-33, Merck, Darmstadt, Germany) for 30 min at RT. After incubation, cells were washed three times for 5 min each with TNTB buffer. Subsequently, cells were incubated with the secondary antibody (Alexa Fluor 488 anti-mouse antibody, 1:100, Jackson ImmunoResearch Europe, Ely, UK) and DAPI (1:100, Thermo Fisher Scientific, Waltham, MA, USA) in TNTB (30 min, RT). Finally, samples were rinsed three times for 5 min each with TNTB buffer, then quickly rinsed with TN buffer (25 mM Tris-HCl, pH ~ 7.5; 150 mM NaCl) and mounted in mounting medium. The fluorescence signals of Alexa Fluor 488 and DAPI were scanned using an Olympus IX81 fluorescence microscope (Olympus, Tokyo, Japan) and evaluated.

### 3.4. Cytotoxicity Assay

The cytotoxicity of FC and FU was assessed using a Cell Count Assay according to [23]. Briefly, the cells were seeded at a density of 5 × 10^3^ cells per well in 96-well plates and incubated for 24 h. Depending on the cells’ doubling times, the tested nucleosides were added to the culture media for either 72 h (HeLa, A549, and hTERT RPE-1 cells) or 96 h (NCI-H2009 and IMR-90 cells). Serial fivefold dilutions of FC or FU were used, starting at a 0.000128 µM concentration and ending at a 50 µM concentration. After this time, the culture media were exchanged for nucleoside-free media for an additional 72 or 96 h. The control wells contained cells without the addition of nucleosides; background wells contained only culture medium.

After the indicated incubation, the culture medium was removed from the wells and cells were fixed with 70% ethanol (RT, 10 min). After fixation, ethanol was removed and samples were air-dried (at least 30 min, RT) and incubated with the 3 µM DAPI solution in 20 mM Tris-HCl, pH 7, and 150 mM NaCl (30 min, RT, 300 rpm). The samples were then quickly rinsed three times for 2 min each with the washing buffer composed of 20 mM citrate buffer, pH 5; 0.5 M NaCl; 0.2% Tween 20; and 2 mM CuSO_4_ (3 × 2 min, RT) and finally with the 20 mM Tris-HCl, pH 7, and 150 mM NaCl. Then, 2% SDS in 20 mM Tris-HCl, pH 7, was added to the wells (15 min, RT, 300 rpm). A volume of 100 µL from each well was transferred to the wells of a black 96-well plate and the DAPI signal was measured using a microplate reader.

### 3.5. Data Acquisition, Computational Approach, and Data Evaluation

When fluorescence microscopy was used, all images were acquired using an Olympus IX83 microscope (UPLFLN 2PH objective 10×), equipped with a Zyla camera (Oxford Instruments Andor, Belfast, Northern Ireland) with a resolution of 1024 × 1024 pixels using acquisition software (Olympus cellSense Dimension 2.3, Olympus, Tokyo, Japan) [25,27].

An Infinite 200 Pro plate reader (Tecan, Männedorf, Switzerland) was used for the measurement of the fluorescence signal in the black 96-well plates. The fluorescence of DAPI was measured at a 370 nm excitation wavelength and 460 nm emission wavelength. The fluorescence of Alexa Fluor 488 was measured at a 488 nm excitation wavelength and 520 nm emission wavelength. The data from the plate reader were analyzed using Microsoft Excel and GraphPad Prism 6 software.

The data analysis was performed using ImageJ 1.54f [28], CellProfiler 4.2.5 (https://cellprofiler.org/ (accessed on 24 April 2023)) [29,30], and Microsoft Excel 2013 software. On average, 10,000 cells were evaluated for every sample if not stated otherwise. The final graphs were generated in GraphPad Prism 6.

The Pearson correlation coefficients, Spearman correlation coefficient, and *p*-values were calculated using the SciPy library [31], GraphPad Prism 6, and SIMCIM tools 1.01 (available from https://privatecloud.imtm.cz/s/QkidNLM473GSmF4) [25]. The Shapiro–Wilk test of normality and Mann–Whitney U test were computed using SciPy library [31] and SIMCIM tools [25].

Adobe Photoshop CS4 was used to prepare the final figures. The scheme in Figure 1 was prepared by Anna Ligasová using draw.io 26.0.3 software. All experiments were conducted in triplicate. The data are presented as mean values ± standard deviation.

## 4. Conclusions

In this study, we developed a novel method for identifying cell lines deficient in cytidine deaminase. This method is based on the conversion of 5-fluorocytidine to 5-fluorouridine and the subsequent detection of its incorporation into RNA. The key steps of the method involve the incubation of cells with 5-fluorocytidine, fixation, permeabilization, incubation with an anti-bromodeoxyuridine antibody that also binds to 5-fluorouridine, subsequent fluorescence detection, image acquisition, and statistical evaluation of the results.

Our findings further indicate that the cytotoxic effects of FU, mediated through its fluorouridine pathway, may contribute to the toxicity observed in healthy cells during FU-based chemotherapy.

Furthermore, we observed a strong, near-perfect proportionality between the FU/FC cytotoxicity ratio and the cellular CDD content. This strong correlation suggests that the primary toxicity of FC can be attributed to its deamination product, FU.

## Figures and Tables

**Figure 1 ijms-26-03344-f001:**
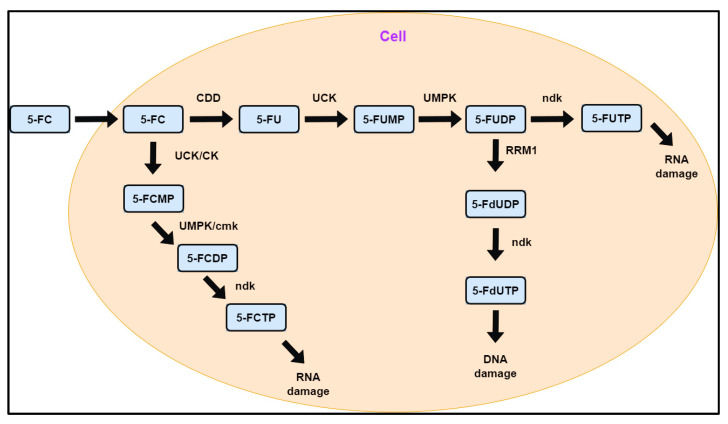
Simplified scheme of the metabolic pathways of 5-fluorocytidine. 5-FC—5-fluorocytidine; 5-FU—5-fluorouridine; 5-FUMP—5-fluorouridine monophosphate; 5-FUDP—5-fluorouridine diphosphate; 5-FUTP—5-fluorouridine triphosphate; 5-FdUDP—5-fluorodeoxyuridine diphosphate; 5-FdUTP—5-fluorodeoxyuridine triphosphate; 5-FCMP—5-fluorocytidine monophosphate; 5-FCDP—5-fluorocytidine diphosphate; 5-FCTP—5-fluorocytidine triphosphate; CDD—cytidine deaminase; UCK—uridine kinase; UMPK—UMP-CMP kinase; ndk—nucleoside-diphosphate kinase; RRM1—ribonucleoside-diphosphate reductase subunit M1; CK—cytidine kinase; cmk—CMP/dCMP kinase.

**Figure 2 ijms-26-03344-f002:**
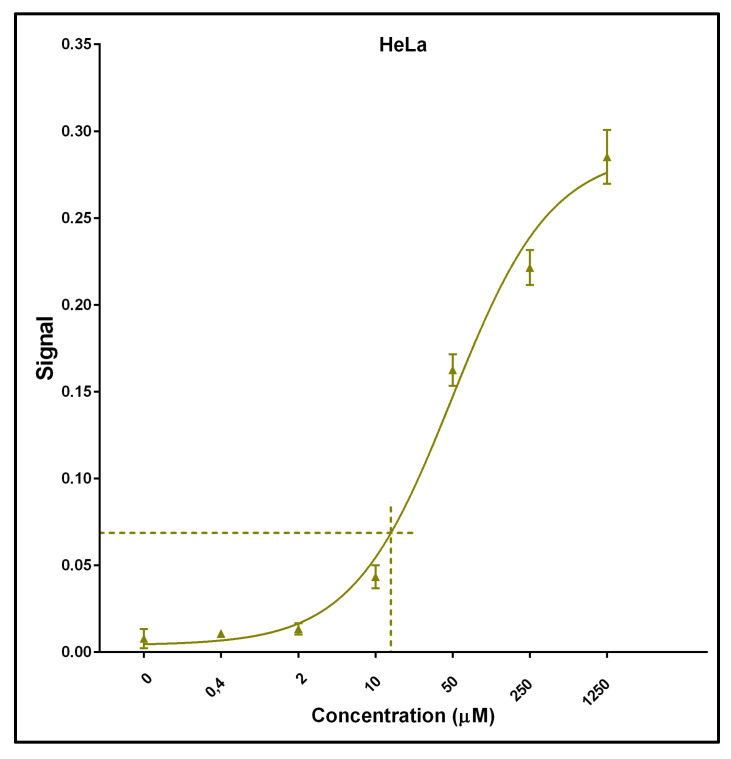
Example of FU calibration curve. HeLa cells were incubated with the indicated concentrations of FU. The dashed line indicates the equivalent FU concentration produced from 10 μM of FC.

**Figure 3 ijms-26-03344-f003:**
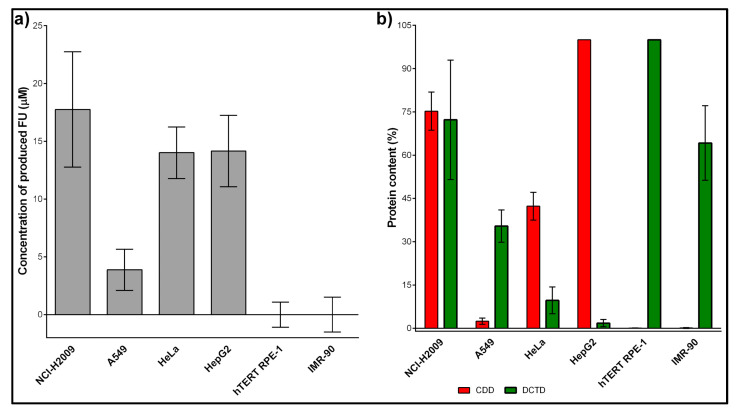
FU incorporation and CDD and DCTD content in various human cell lines. (**a**) FU incorporation after the deamination of FC by CDD in various cell lines is shown. The data are presented as the mean ± SD. (**b**) CDD and DCTD content determined by Western blot is shown. The data are presented as the mean ± SD. Panel (**b**) is adapted from Figure 4 of [20].

**Table 1 ijms-26-03344-t001:** Statistical analysis of FU/FC treatment effects (Mann–Whitney U test; *p*-values reported).

	Ctrl vs. 50 µM FU	Ctrl vs. 50 µM FC
NCI-H2009	0.00214	0.00269
A549	0.00054	0.00028
HeLa	0.00003	0.00012
HepG2	0.00084	0.00002
hTERT RPE-1	0.00152	0.16752
IMR-90	0.00297	0.54762

**Table 2 ijms-26-03344-t002:** Correlation between FU/FC toxicity ratio and protein levels.

	Pearson Coefficient	*p*-Value
DCTD	−0.678	0.14
hENT1	−0.143	0.79
hENT2	−0.729	0.10
CNT1	−0.105	0.84
CNT2	−0.671	0.15
CNT3	−0.503	0.31

## Data Availability

All data are included in this article (and Appendix A). Additional data supporting the findings are available from the corresponding authors upon reasonable request. The SIMCIM tools software can be downloaded from https://privatecloud.imtm.cz/s/QkidNLM473GSmF4.

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
