# Peer review of "A Rapid Approach for Identifying Cell Lines Lacking Functional Cytidine Deaminase"

_ijms, 2025, doi:10.3390/ijms26073344_

Round 1

Reviewer 1 Report

Comments and Suggestions for Authors

A manuscript of Ligasova et al describes an elegant protocol for detection of cytidine deaminase deficient cell lines.

This is a solid manuscript describing a quantitative approach for identification of cytidine deaminase activity across the cell lines. Proposed approach can enable a prognostic analysis of cellular sensitivity to cancer drugs. Potentially, it may support studies of cellular heterogeneity and could become popular tool in the field. In general, the manuscript is well written an I suggest it is accepted for publication with the following improvements.  

1) The physiological role of CDD should be described both in Abstract and Introduction.

2) Considering that CDD is a housekeeping enzyme required for uridine production it is an interesting fact that some cell lines can survive without it. If any alternative mechanism for uridine synthesis is known please discuss this information.

Author Response

Thank you very much for your insightful comments and suggestions. We have carefully considered your feedback and made the following changes to the manuscript to address your points:

We have added the sentence "CDD plays a pivotal role within the pyrimidine salvage pathway" to the Abstract. It can be found on Page 1.

We have also modified the Introduction section to include the additional details regarding the role of cytidine deaminase. In the Introduction, we also mentioned an alternative pathway for uridine synthesis. These changes are also located on Page 1.

We believe that these revisions have significantly improved the manuscript, and we appreciate your time and expertise in helping us to enhance the quality of our work.

Reviewer 2 Report

Comments and Suggestions for Authors

Anna Ligasová et al describe new method for identification of cell lines lacking functional cytidine deaminase 

The description of the immunocytochemical detection method is concise, yet it conveys the essential details needed for understanding the approach.

The correlation between the 5-fluorouridine/5-fluorocytidine cytotoxicity ratio and cytidine deaminase content is clearly stated, reinforcing the significance of the method.

------------------------------------------------------------------------

1) Abstract need improve it .The description of the method is clear, but specifying how immunocytochemical detection is performed would improve clarity for readers 

2) Also rewrite completely conclusions and not divide in sections 4.1.  and 4. 2.

3) other :

in abstract : The long list of non-correlating factors ("deoxycytidine monophosphate deaminase content, or equilibrative nucleoside transporters 1 or 2, or concentrative nucleoside transporters 1, 2, or 3.") could be formatted more clearly. Consider breaking it into a separate sentence or using bullet points if appropriate in the full document.

Hence , this reviewer indicate major revisions for publication of paper in IJMS.

Author Response

Thank you for your valuable feedback and suggestions for improving our manuscript. We have carefully considered your comments and made the following revisions:  

Abstract: We have improved the clarity of the abstract by specifying how immunocytochemical detection was performed and by dividing the long list of non-correlating factors into separate sentences.

Conclusions: We have rewritten the Conclusions section to integrate the information from the subsections into a single, cohesive conclusion.

We believe that these changes have significantly enhanced the manuscript's clarity, accuracy, and overall quality. We appreciate your insightful comments and the time you have dedicated to reviewing our work.